# Organochlorine Pesticides and Salinity in Karakalpakstan, Uzbekistan: Environmental Health Risks Associated with the Aral Sea Crisis

**DOI:** 10.3390/ijerph22111751

**Published:** 2025-11-19

**Authors:** Casey Bartrem, Murad Ismaylovich Kurbanov, Brock Daniel Keller, Andrea Fiori, Ian von Lindern, Polat Zoldasbaevich Khajiev, Dilmurod Rustamov, Jerry Lee, Marina Steiner, Zamira Paluaniyazova

**Affiliations:** 1TerraGraphics International Foundation, Moscow, ID 83843, USA; brock.keller@duke.edu (B.D.K.); ian@terrafound.org (I.v.L.); jblee2017@gmail.com (J.L.); 2Department of Soil and Water Systems, University of Idaho, Moscow, ID 83843, USA; marina@uidaho.edu; 3Ministry of Health of the Republic of Karakalpakstan, Nukus 230100, Uzbekistan; kurbanovm1972@gmail.com (M.I.K.); mambetmuratovapikon@gmail.com (P.Z.K.); zarina838409@gmail.com (Z.P.); 4Médecins Sans Frontières, Tashkent 100070, Uzbekistan; andrea.fiori@moscow.msf.org (A.F.); rdilmurad@gmail.com (D.R.)

**Keywords:** Uzbekistan, Aral Sea, environmental health, pesticides, water quality, salinization

## Abstract

The legacy of Soviet-era agricultural practices in Central Asia has contributed to severe environmental degradation through residual organochlorine pesticide contamination, dramatic reduction in surface water, and the near-total desiccation of the Aral Sea. Few studies have investigated hazards to human health, despite the region’s elevated burden of disease. This study aimed to characterize environmental hazards in the Republic of Karakalpakstan, one of the most environmentally and economically impacted regions. Environmental assessment included the collection of 140 soil, water, and sediment samples across 79 unique locations. Pesticide results showed organochlorine pesticides over US reference levels in 100% of water samples, with 30% also exceeding in hexavalent chromium. Water salinity is a primary concern: expressed as total dissolved solids, values ranged from 563 to 3852 mg/L. Over half of the 46 soil and sediment samples tested above reference levels for aldrin. Soil and sediment sample salt content reached up to 8.7%. Residual persistent organochlorine pesticides remain a significant health risk in Karakalpakstan, while water availability is decreasing, and water quality parameters, including salinity, dissolved oxygen, pH, dissolved metals, and nitrate levels, are degrading. Such challenges extend beyond the Aral Sea Basin: as salinization and desiccation of endorheic lakes continue on a global scale, similar situations may become commonplace. Research and interventions from this region can serve to support other similarly impacted areas.

## 1. Introduction

The desiccation of the Aral Sea is considered one of the worst ecological disasters in recent history. The Soviet Union’s “hydraulic mission” in Central Asia saw mass irrigation projects designed to support cotton agriculture [1]. By the end of the 20th century, nearly 75% of cultivated land in the hydrologic basin was designated for cotton production [1]. As a result, 96% of all water flow from the Amu Darya and the Syr Darya Rivers feeding the sea was lost to irrigation [2]. Once the world’s fourth-largest inland body of water, the Aral Sea has diminished to 9% of its historic volume [3]. Climate change and upstream river diversions by other countries further deplete the flow. The Amu Darya no longer reaches the Aral Sea, now terminating in the Kiziljar lagoon, 27 km from the former seaport of Moynaq.

Soviet-era cotton production required intense application of organochlorine pesticides, including aldrin, lindane (γ-hexachlorocyclohexane [HCH]), and dichlorodiphenyltrichloroethane (DDT), which persist in regional soil and water at alarming concentrations [4]. The Aral Sea served as a natural repository for these pesticides and agricultural byproducts [5]. Meteorological conditions over the former seabed result in increasingly frequent dust storms that transport residual contaminants up to 500 km, representing a novel health risk for the region’s population [6,7]. Increasing soil salinity has become a growing problem as water resources decline, threatening regional agriculture [2]. In response, the practice of field flooding to remove salt content and subsequent application of nitrogenous fertilizers to account for losses in soil nutrients and humic material have become commonplace, exacerbating salinity and geochemical changes as these are recycled into the water supply [8]. Drinking water salinity can be up to six times greater than World Health Organization standards [9], and more than half of Karakalpakstan’s irrigated lands are characterized as saline or very saline [2]. Excessive fertilizer application contributes to the alteration of soil and runoff water mineral saturation and physical properties that compound soil erosion and salinity [10].

The Karakalpak hydrologic system, and subsequently the regional economy, depends on the management of the Amu Darya. The entire river flow is diverted to massive canal systems to irrigate reclaimed desert lands. Groundwater sources are replenished by seepage from the irrigation or return flow collector canals. As a result, water quality is largely dependent on the Republic’s agricultural practices.

Due to its proximity to the dry seabed, prevailing northerly winds, and dependence on Amu Darya River diversion for drinking water, the Republic of Karakalpakstan stands out as one of the more severely affected regions of the Aral Sea disaster [11,12]. High incidence of non-communicable diseases, including anemia, kidney and liver disease, respiratory infections, allergies, and cancer, has long been recorded [5,13,14,15,16,17,18,19]. Between 2002 and 2005, a series of collaborative national and international investigations and intervention activities were conducted. Studies by Ataniyazova, Madreimov, and co-authors provided notable descriptions and accounts of disease patterns, etiology, and exposure pathways, detailing the relation between environmental degradation in the Aral Sea region and human health outcomes. These studies highlighted health concerns for vulnerable groups, especially women and children. Few of these surveys, however, characterized specific environmental factors likely responsible for these health concerns [10]. Despite the evident health crisis, most investigations were curtailed around 2005 when investigations and international collaborations were discouraged by Uzbek authorities. Consequently, reports of environmental health-related research in Karakalpakstan are scarce in the literature from 2005 to the present. Although various government ministries continued to collect health and environmental quality information following the decline in international collaboration, comprehensive summaries or analyses of health and environmental quality are uncommon and difficult to obtain.

In response to the need for an updated, comprehensive environmental assessment to inform public health interventions, a multidisciplinary investigation was initiated by the Ministry of Health of the Republic of Karakalpakstan (MOHK), national and rayon-level branches of the Sanitary and Epidemiological Service (SES), Médecins Sans Frontières (MSF), and TerraGraphics International Foundation (TIFO). This study aimed to further characterize the concentrations and prevalence of three environmental contaminants in the sources identified by local researchers and health authorities, including legacy pesticides, salinity, and byproducts of fertilizer amendments in soil, sediment, water, and air. Environmental samples were collected over a four-week period during the months of March and April 2023, covering most major population centers and agricultural regions of Karakalpakstan, with special attention to locations where women and young children are more likely to be exposed to contaminants. By establishing a baseline understanding of environmental conditions, this study seeks to provide essential insights for future public health intervention efforts in the Aral Sea region.

## 2. Materials and Methods

Sampling occurred from March–April 2023 within 6 zones, detailed in Figure 1, which were defined by location along the southern (upstream) and northern (downstream) sections of the Amu Darya River and its irrigation canals. The zones were delineated to encompass the majority of the populated and agricultural regions and allowed a focused region for passive air sampling within the study timeframe. Passive sampling results are not presented in this report. Areas outside of these zones are largely desert with a sparse population. Table 1 shows the number and type of samples collected by zone.

A total of 140 samples were collected at 79 sampling locations, including 70 water, 54 soil, and 16 sediment samples. Sample locations and types were selected to be most relevant for public health: playgrounds, schools, agricultural areas, and water supply systems. Water and soil sample locations included schools, playgrounds, residences, orchards, and agricultural fields. Split samples were collected to enable intra-laboratory data comparison.

Surface water, groundwater, and tap water samples were collected in 1 L glass containers. Containers were thoroughly washed and rinsed with deionized water prior to use. Water samples were obtained from government water treatment plants, community wells, irrigation canals, and agricultural runoff “collector” canals. Sampling procedures followed the US EPA 817-R-08-003 Guidance for Unknown Contaminants in Drinking Water [20]. Samples were not acidified prior to refrigeration to ensure the integrity of pesticide and salinity results.

Soil samples were collected with stainless steel implements at schools, playgrounds, residences, orchards, and agricultural fields. All soil samples were composite samples collected based on US EPA 540/R-95/141 and included a minimum of 5 equally portioned aliquots that were homogenized in stainless steel bowls before being transferred to glass containers [21]. Sediments were collected from the Amu Darya River and canals. Sediment samples were collected, homogenized using stainless steel implements, and transferred to glass containers per EPA method FSBPROC-200-R6 [22].

All samples were collected in sterilized glass jars and stored at 4–8 °C for the duration of the sampling period (4 weeks), until delivery to laboratories immediately following the conclusion of the sampling effort. In the field, samples were kept at 4–8 °C in coolers with ice, and periodically transferred to refrigerators in Nukus during supply restocks. Samples for organic analyses were collected in amber borosilicate glass containers.

Public areas with intensive use by young children (e.g., schools and playgrounds) were screened for soil heavy metals in situ with a handheld X-ray fluorescence spectrometer (XRF) (NitonTM XL3t GOLDD+, ThermoFisher Scientific, Waltham, MA, USA).

Samples were distributed among three national laboratories for analysis to accommodate quality control assurance requirements and to provide an array of technical data reflecting laboratory analytical capacity, as summarized in Table 2. Organochlorine and organophosphorus pesticide screening was conducted using gas chromatography mass spectrometry in an Agilent 5977A machine (Agilent Technologies, Santa Clara, CA, USA) following the IU 012-3/0010 methodology. Iron, copper, fluorine, sulfates, hexavalent chromium, nitrates, and nitrites were measured in water samples by spectrophotometry in a V-1200 device (VWR International, Radnar, PA, USA) [23], while in soils, nitrites and nitrates were measured by an Expert 001 ionomer device (EasyLife, Bishkek, Kyrgyzstan) [24]. Chlorides and total hardness were determined by titrimetric analysis, and dry residue by weight [25,26,27].

Salinity measurements at the AgroChemical laboratory were reported by deriving the individual concentrations of specific salt ions according to the Uzbek national agrochemical OST46-52-76 “Methods of agrochemical analysis of soils. Determination of chemical composition of water extracts and groundwater composition for saline soils”. These measurements are based on the extraction of soluble salts from a 1:5 soil–water slurry, followed by the determination of the ions: CO_3_^2−^, HCO_3_^−^, Cl^−^, Ca^2+^, Mg^2+^, Na^+^, K^+^, SO_4_^2−^. Methodologies for ion quantification are described in Table 3.

In addition to the environmental samples, three PurpleAir© particulate monitors (PurpleAir Inc., Draper, UT, USA) were permanently installed and continue to actively report live air quality data to the publicly accessible PurpleAir© network. Two are located at the Rayon SES buildings in Mangit and Moynaq, with the third located at the MOHK Nukus office.

During sampling, field staff recorded qualitative observations of water treatment and agricultural practices. While these observations were not part of the quantitative sampling protocol, they provide context for potential public health risks. Relevant observations are included in the Section 4.

## 3. Results

### 3.1. Quality Assurance and Quality Control

Results from double-blind soil and water samples analyzed at the Tashkent SES laboratory were generally comparable (see Appendix A). Water results for α-HCH and β-HCH were moderately different but within the error values for the reporting laboratory. Results for lindane, DDT, and its byproducts dichlorodiphenyldichloroethylene (DDE) and dichlorodiphenyldichloroethane (DDD) were all comparable. Soil results were comparable for β-HCH, lindane, and heptachlor. However, results for α-HCH and aldrin were several orders of magnitude different. Additionally, dieldrin, a metabolite of aldrin, was not detected in any of the samples. These unexpected results call the reliability of α-HCH and aldrin into question; they should be investigated in follow-up investigations. Double-blind sample results from the Nukus SES laboratory and the AgroChemical laboratory were all comparable.

### 3.2. Water Results

Pesticide analysis in water samples sent to the Tashkent National SES laboratory revealed elevated concentrations of organochlorine pesticide byproducts. Lindane was detected in most samples, although none exceeded the US Environmental Protection Agency (EPA) Maximum Contaminant Level (MCL) of 0.2 µg/L. However, lindane isomers α-HCH and β-HCH, degradation byproducts of legacy lindane applications, were prevalent and frequently greater than EPA regional screening levels (RSLs) (0.0072 µg/L and 0.025 µg/L, respectively). Forty-nine (49) of the fifty water samples contained concentrations of the alpha-isomer above the RSL. α-HCH detections ranged from 0.02 to 3.7 µg/L, with nearly half of the samples an order of magnitude greater than the RSL. The beta-isomer was detected in every sample, with 45 of the 50 having concentrations above the RSL. β-HCH detections ranged from 0.01 to 0.8 µg/L. In total, 90% of all water samples tested above the RSLs for both the alpha and beta lindane isomers.

Residual DDT levels were below the RSL of 0.23 µg/L, despite the fact that application rates from 1980 to 1992 reached 72 kg/hectare in RK (compared to 1.6 kg/hectare in the US and 4 kg/hectare in Russia) [11]. However, DDT’s persistent, bioaccumulative degradation byproducts DDE and DDD were both frequently detected, with some exceedances of the RSLs (0.046 and 0.032 µg/L, respectively). DDE screening recorded eight exceedances, two of which were greater than one order of magnitude above the RSL, with concentrations of up to 1.26 µg/L. Only a single sample exceeded the DDD RSL.

Table 4 summarizes pesticide results for 50 water samples. Each of the 50 total water samples surpassed the RSL for at least one of the three predominant organochlorine pesticides (α-HCH, β-HCH, and DDE), and 7 samples showed concentrations above the RSLs for all three contaminants. Of the samples exceeding all three RSLs, five were from community wells, or approximately 25% of all well samples.

Table 5 summarizes primary water quality parameters. Six (6) of the 20 water samples analyzed at the Nukus SES Laboratory contained elevated levels of Cr^6+^. Though none of the samples exceeded the 0.1 mg/L MCL for Cr^6+^, six did exceed the RSL of 0.000035 mg/L. The Nukus SES laboratory did not observe any MCL exceedances for nitrates or nitrites.

Table 6 summarizes water salinity estimates for 20 water samples, using total dissolved solids (TDS) as a proxy. Values varied considerably between water sources but were consistently elevated. Surface water TDS levels ranged from 563 mg/L to 1766 mg/L, with most observations in the 1200 mg/L to 1400 mg/L concentration range. Similar TDS levels were observed in the southern and central reaches of the Amu Darya. Well water sources had the greatest fluctuation in overall TDS, with those in the most southern zone containing notably lower TDS concentrations, near 800 mg/L. Well water sources farther north, however, showed higher salinity, ranging from 1600 mg/L to 1932 mg/L TDS. Collector canals show the highest TDS concentrations, typically exceeding 2500 mg/L to >3500 mg/L, as expected.

### 3.3. Soil and Sediment Results

Table 7 summarizes soil and sediment pesticide results. Persistent pesticides were less prevalent in soils and sediments than those observed in water. No exceedances for lindane and its byproducts, or DDT and its byproducts, were found in soil or sediment samples. Aldrin was detected in more than half of the samples analyzed for pesticides at Tashkent SES, at concentrations exceeding the RSL of 0.039 mg/kg. Aldrin readings ranged from 0.0001 to 3.753 mg/kg. The absence of dieldrin is unusual, and the cause of which merits further investigation. Heptachlor was the only other pesticide discovered at high concentrations, with two exceedances of the 0.13 mg/kg RSL.

Table 8 summarizes soil and sediment salinity results. Soil salinity tended to correspond to soil maintenance practices. Recently flushed agricultural fields and home gardens yielded the lowest salinity levels, while schoolyards and playgrounds revealed higher levels, possibly because these are not routinely flushed. As expected, the highest salinity levels were observed in canal sediments. Following the Karakalpak system for ranking salinity concentrations, all the soil and sediment samples were classified as “high” or “very high” salinity and were each accorded one of four major “salting types” corresponding to the ratio of sulfate to chlorine ions in milliequivalents. Most soil samples were sulfate-dominated. Soil salinity levels were stratified geospatially: all soil samples from the southernmost (upstream) zone were classified as “saline”, while samples from northern (downstream) zones typically showed higher salinity levels. Five (5) of the twenty-four samples exhibited magnesium-induced soil degradation, instances in which Mg^2+^ accounts for >30% of the cation exchange complex.

Results from in situ XRF screening did not indicate any concerns for heavy metal toxicity. Several samples showed slightly elevated concentrations of Co, Se, and Zr.

### 3.4. Air Quality Results

Six months of air quality index (AQI) data from the PurpleAir© monitors do not suggest significant respiratory risk from fine particulate matter (PM). Elevated concentrations of 10 µm, 5 µm, and 2.5 µm PM are noted periodically and tend to follow typical diurnal anthropogenic patterns or windstorms. Persistent high-level readings are uncommon, although the significance of these data is limited as the monitors have not been in place long enough for potential seasonal effects to be evaluated.

## 4. Discussion

These results are an updated snapshot of environmental conditions that complement a 2018 publication by the Samarkand branch of the Uzbek Academy of Sciences, which published a monograph of analyses of the preceding 15 years of Karakalpak health and environmental monitoring data and improved understanding of the relationships between disease incidence and environmental quality in the Aral Sea region. Researchers found disease and overall morbidity rates are increasing, and specifically identified associations of disease in blood and blood-forming organs with water quality [5,17], functional relationships between rates of respiratory diseases and the level of air pollution [13,16,18], and relationships between the incidence of malignant neoplasms and contaminated water samples and atmospheric air [28]. Additional diseases were elevated compared to other regions, including bronchopulmonary pathology, astigmatism, impaired immune system function, increased heterochromism, and congenital abnormalities among children, as well as allergic diseases among children [5,9,10,12,14,15,16,18,29,30,31]. This is aligned with findings from older investigations that found 17 dioxins, including high levels of the extremely toxic 2,3,7,8-Tetrachlorodibenzo-p-dioxin, in human breast milk [32].

Results from our study are the first comprehensive survey in decades to compare chlorinated compounds and other parameters to health-based drinking water standards in Karakalpakstan. Results suggest that environmental exposures are likely contributing to the elevated rates of disease identified by other researchers in Karakalpakstan and in Kazakhstan [19,33]. Considering the variability in water flows and salinity, the magnitude and complexity of the water diversion systems, and changing climate patterns, all of which contribute to increasing water scarcity, coupled with the seasonality of agricultural chemical application [34], these results provide a snapshot in time; additional assessments during different seasons and over the course of multiple years would provide a more comprehensive understanding. However, the results offer an essential baseline for future monitoring and epidemiologic studies, identify compounds of concern warranting follow-up investigation, and clearly confirm the presence of hazardous concentrations of persistent pesticides in the environment.

Monitoring for pesticides in the environment by Karakalpak agencies is limited; they have continued to routinely monitor residual DDT, HCH, and byproducts. However, analyses are conducted by thin-layer film chromatography and titration techniques and have shown few detections of these analytes in the past decade (data under review and not available at the time of publication). The agencies are in the process of updating analytical laboratory capabilities, but at the time of this study (2023), there was no laboratory capacity in Karakalpakstan to conduct modern organochlorine, organophosphorus, or organonitrogen chemical screens. The results obtained in this study from the National SES Laboratory in Tashkent are among the first comprehensive analyses of drinking water contamination using state-of-the-art equipment.

While our study area was restricted to RK, it is probable that similar environmental conditions and public health concerns prevail throughout other parts of the greater Aral Sea Basin. Evidence of elevated dieldrin, lindane, and lindane-derivative concentrations has been recorded in the neighboring Kazakh Syr Darya river system just downstream of Uzbekistan’s populous and agriculturally intensive Ferghana Valley [35].

Declining water flows, poor agricultural soil conditions, increasing water salinity levels, deteriorating drinking water infrastructure and disinfection facilities, and irregular irrigation water supplies adversely affecting shallow wells used by rural populations all exacerbate public health risks. These factors, combined with adverse socio-economic, nutrition, and poverty-related risk co-factors, likely contribute to the region’s elevated burden of disease. The implications of agricultural practices, drinking water quality, and vulnerable populations are discussed in the following subsections.

### 4.1. Agricultural Practices and Chemical Amendments

The Aral Sea Basin is reportedly the northernmost major cotton-producing region in the world, and one of the few relying on annual cropping [36]. To maintain productivity, the fields are flooded in the off-season to reduce excess alkaline salts, contributing to ever-increasing salinization of groundwater and surface waters as overall water supplies decrease [7]. Residual pesticides and nutrients flushed with the salts contribute to the consistently elevated contamination of surface, groundwater, and drinking water supplies observed during the 2023 assessment.

Despite the widespread use of agricultural soil amendments in the study area, nitrate and nitrite concentrations were lower than anticipated. This likely reflects the timing of our sampling rather than an absence of nutrient inputs. Sampling occurred in March–April 2023, prior to the main period of fertilizer and amendment application in the region. Consequently, our findings represent pre-application baseline conditions, which may explain the lack of elevated nitrate and nitrite levels observed. In contrast, the detection of historic pesticide byproducts at concerning levels underscores the persistence of legacy contaminants in the system. Future sampling later in the planting season would likely capture higher concentrations of nitrates and nitrites associated with active agricultural inputs, providing a more complete picture of seasonal contaminant dynamics.

Given the prevalence of agricultural soil amendments, nitrate and nitrite readings were unexpectedly low. This study prioritized the preservation of pesticides and TDS over other contaminants, and, as such, samples were not acidified. Storing water samples at a pH of <2 would likely enhance nitrate and nitrite recovery (see EPA method 300.0, Revision 2.1, Section 8.2), though the lack of elevated readings may also be explained by the timing of the sampling relative to the season when nitrate and nitrite compounds are being applied. The timing of our study in March–April 2023 likely preceded the period when soil amendments are applied. Thus, toxic compounds other than those identified in this study may be contributing to the observed disease prevalence.

### 4.2. Drinking Water Infrastructure/Water Quality

The Karakalpak water supply and distribution system is complex and highly variable. A 2017 study by the Asian Development Bank (ADB) provided a comprehensive description of the system and summarized water quality data along the Amu Darya at key diversion points servicing the population’s drinking water supply for the 2016 water year [13,34]. Approximately 40% of the population received drinking water from the centralized supply. These waters are sourced from five major irrigation canals diverted from the Amu Darya to three water treatment plants and two headworks that provide primary settling and chlorination. The TDS and chemical content of the water entering the distribution system largely reflect conditions at the river diversions [3].

Comparison of the ADB results to the 2023 data at the same location and season suggests alarming trends. In 2023, dissolved oxygen levels in the Tuyamuyun reservoir, where the Amu Darya enters RK, were less than half of the minimum value observed in 2016, while TDS, pH, nitrate, iron, and copper levels were all higher than the maximum levels observed in 2016. The Tuyamuyun reservoir is the main diversion point for a 300 km long pipeline providing drinking water for a substantial portion of the Karakalpak population [24]. The TDS results from this March–April 2023 sampling period correspond with the temporal peak in the 2016 data, but 2023 salinity levels at downstream drinking water diversion points were 30% to nearly 50% higher, reflecting declining water quality of the Amu Darya as it enters and flows through Uzbekistan towards the remnants of the Aral Sea.

High salinity and chemical composition of water throughout the system suggest the potential for a “salinity cocktail” effect, where interactions between chlorine, salt ions, and chemical contaminants in the water may increase leaching of, or facilitate chemical reactions that produce, more toxic compounds [37]. This can be a particular concern in waters heavy in organic materials, nitrogen compounds such as fertilizer amendments, or pesticides. The widespread potential for such interactions merits more in-depth investigation, especially where water sanitation protocols seem to vary between the water treatment plants and distribution system headworks, as these differences can have direct implications for a “salinity cocktail” effect. Field observations showed that local water supplies may be subject to periods of over-chlorination or no chlorination. These results indicate potential health concerns for the larger urban population not identified in studies to date. Future sampling protocols should include characterization of both inorganic nitrogen and organonitrogen compounds in drinking water to better understand the extent of nitrogen contamination from agricultural additives and implications for human health.

### 4.3. Vulnerable Populations

The ADB reported that about 10% of rural dwellers rely on known unsafe supplies that tap intermittent perched water adjacent to irrigation or collector canals [3]. Samples from these shallow wells in 2023 were consistently above the median values for pesticide residues and included the highest values for both α- and β-HCH, supporting ADB’s findings. Several rural residents described concerns that many of these wells are failing, and reliability depends on concurrent canal flows, indicating sources are dependent on irrigation water delivery and associated canal seepage.

## 5. Conclusions

This study was a multilateral collaboration that aimed to better understand the environmental and health risks associated with deteriorating water quantity and quality in Karakalpakstan. The investigation complements and supports previous Karakalpak research correlating excess disease incidence in the region to deteriorating environmental conditions.

Results demonstrate that residual Soviet-era organochlorine pesticides remain a significant health concern, as most of the water samples from this study exceeded Uzbek and international reference levels throughout the study area. While moderate levels of pesticide residues were detected in many soil and sediment samples, the consistently elevated presence of salts and byproducts of lindane and DDT in drinking water likely represent direct environment-based health threats. Increasing salinity, combined with the presence of organonitrogen compounds and irregular chlorination practices, may enhance the toxicity of contaminants being delivered in the centralized drinking water systems. Much of RK’s rural population is without access to safe water, and future investigations should better characterize these populations’ potential exposures, as poverty-related, nutrition, endemic disease, and limited access to healthcare enhance vulnerability to adverse health effects.

Importantly, although the Aral Sea is an extreme example, desiccation of endorheic lakes, salinization, and the associated adverse human and environmental health consequences are not unique to RK. Climate change, irregular water supply, declining water quality, increasing dependence on agricultural chemicals, water management infrastructure, salinization of drinking water, and provision of safe drinking water are pressing global issues. The RK situation may be a harbinger of a resource management crisis that, left unmitigated, may evolve as among the more important climate and human health challenges of this century.

## Figures and Tables

**Figure 1 ijerph-22-01751-f001:**
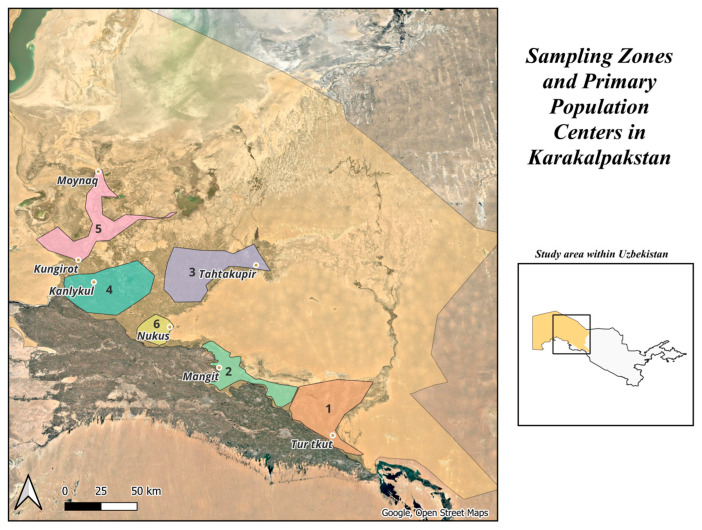
Sampling zones 1 through 6 are indicated in colored polygons within the 2023 study area. Primary population centers are also indicated within each of the six sampling zones. The remnant of the Aral Sea is visible in the northern portion of the map. The irrigated area of the region is visible in the green areas, while the remainder of the region is desert. The center of Karakalpakstan is located at 43.7° N 59.0° E, and Nukus is at 42.47° N and 59.61° E.

**Table 1 ijerph-22-01751-t001:** Number and type of samples by zone during the March–April 2023 sampling. Water samples were collected from government water treatment plants, community wells, irrigation canals, and agricultural runoff “collector” canals.

Zone	Water Samples	Soil Samples	Sediment Samples	Total Samples by Zone
Zone 1	15	6	5	26
Zone 2	10	6	2	18
Zone 3	9	12	2	23
Zone 4	12	5	4	21
Zone 5	11	13	2	26
Zone 6	13	12	1	26
Total samples by type	70	54	16	140

**Table 2 ijerph-22-01751-t002:** Division of samples by laboratory and analytics.

Lab	Water Samples	Soil Samples	Sediment Samples	Analysis
Tashkent National SES Laboratory	50 (71%)	35 (65%)	11 (69%)	Organochlorine, organophosphorus, and organonitrogen compound screening
Nukus Republican SES Laboratory	20 (29%)	0	0	Standard Drinking Water and Open Reservoir analysis
AgroChemical Analysis Center of the Republic of Karakalpakstan	0	19 (35%)	5 (31%)	Salinity and metals analysis

SES = Sanitary epidemiological station.

**Table 3 ijerph-22-01751-t003:** Salt ions and corresponding AgroChemical laboratory analysis methodology.

Ion(s)	Methodology
CO_3_^2−^ and HCO_3_^−^	Determination of CO_3_^2−^ and HCO_3_^−^ ions (carbonate and bicarbonate alkalinity) by successive titration with sulfuric acid solution
Cl^−^	Mercurimetric method for determination of Cl^−^ ion
Cl^−^	Mohr argentometric method for determination of Cl^−^ ion
Ca^2+^	Trilometric method for determination of Ca^2+^ ion (based on titration of calcium ions with Trilon B in a strongly alkaline medium in the presence of murexide as a metal indicator)
Ca^2+^ and Mg^2+^	Trilometric method for the determination of the sum of Ca^2+^ and Mg^2+^ ions
Na^+^ and K^+^	Flame-photometric method for definition of Na^+^ and K^+^ ions
SO_2_^4−^	Weight method for determination of SO_2_^4−^ ion (based on precipitation of sulfate ion with barium chloride and weighing the calcined precipitation as BaSO_4_)

**Table 4 ijerph-22-01751-t004:** Primary water pesticide summary from the Tashkent National Laboratory (*n* = 50).

Parameter	US EPA Residential RSL (µg/L)	Range (µg/L)	Median (µg/L)	First Quartile	Third Quartile	Surface Water RSL Exceedances	Well Water RSL Exceedances	Collector Canal RSL Exceedances
α-HCH	0.0072	0–3.7	0.07	0.0525	0.123	96%	100%	100%
β-HCH	0.025	0.01–0.8	0.07	0.042	0.1175	91%	86%	100%
Lindane (γ-HCH)	0.042	0–0.049	0.01	0.004	0.02	2%	0%	0%
DDE	0.046	0–1.258	0.01	0	0.01	13%	23%	0%
DDD	0.032	0–0.05	0.01	0.003	0.0175	4%	0%	0%
DDT	0.23	0–0.02	0	0	0.01	0%	0%	0%

**Table 5 ijerph-22-01751-t005:** Primary water quality parameters from the Nukus Republican SES Laboratory (*n* = 20). US EPA RSLs and MCLs are provided for comparison, with MCL values indicated by ^†^. N/A = not applicable, LOD = limit of detection.

Parameter	Units	US EPA RSL or MCL ^†^	Range	Median	First Quartile	Third Quartile
Turbidity	mg/L	N/A	0.0–24	0.3	0.2	2.4
pH	-	N/A	7–8.1	8.0	7.775	8
Oxidability (dissolved oxygen)	mg O_2_/L	N/A	0.3–2.2	0.750	0.4825	1.2
Total hardness	meq/L	N/A	0.3–33.7	11.6	9.95	15.2
Nitrate	mg/L	10 ^†^	0.1–7.1	2.7	0.675	4.325
Nitrite	mg/L	1	0.007–0.5	0.007	0.007	0.013
Hexavalent chromium	mg/L	0.0000035	<LOD–0.04	<LOD	0	0.008
Dry residue	mg/L	N/A	563–3852	1561	1311.75	1807.5
Chlorides	mg/L	Chlorine ^†^: 4 Chlorite ^†^: 1	149–1046	252	231.5	340.75
Sulfates	mg/L	N/A	48–345	256.5	219	288
Fluorine	mg/L	4 ^†^	<LOD–0.6	0.09	0.06	0.13
Copper	mg/L	1.3 ^†^	0.02–0.4	0.17	0.095	0.265
Total iron	mg/L	N/A	<LOD–0.4	0.087	0.07	0.13

**Table 6 ijerph-22-01751-t006:** Water salinity summary from the Nukus Republican SES Laboratory (*n* = 20).

Parameter	Range (mg/L)	Median Value (mg/L)
Surface water TDS	563–1766	1356
Well water TDS	793–1932	1561
Collector canal water TDS	1420–3852	2534

**Table 7 ijerph-22-01751-t007:** Soil and sediment pesticide results from the Tashkent National SES Laboratory compared to the US EPA regional screening level (RSL) values (*n* = 46).

Parameter	US EPA Soil RSL (mg/kg)	Range (mg/kg)	Median Value (mg/kg)	First Quartile	Third Quartile	Quantity of RSL Exceedances
α-HCH	0.086	ND *–0.0231	0.0001	0.0000425	0.000125	0
β-HCH	0.3	ND–0.054	0.0001	0.00003175	0.000183	0
Lindane (γ-HCH)	0.57	ND–0.008	0	0	0.00002	0
Heptachlor	0.13	ND–0.2197	0.0018	0.0001	0.010488	3 (7%)
Aldrin	0.039	ND–3.752	0.046545	0.01109	0.10475	25 (54%)
DDE	2.0	ND–0.23274	0	0	0.00001	0
DDD	2.3	ND–0.0062	0	0	0	0
DDT	1.9	ND–0.38019	0	0	0.00001	0

*ND = non-detect. Tashkent SES did not provide a detection limit for the results.

**Table 8 ijerph-22-01751-t008:** Soil and sediment salinity summary from the AgroChemical Analysis Center of the Republic of Karakalpakstan (*n* = 24).

Parameter	Zone 1 Range (% Salt)	Zone 2 Range (% Salt)	Zone 3 Range (% Salt)	Zone 4 Range (% Salt)	Zone 5 Range (% Salt)	Zone 6 Range (% Salt)
Agricultural Field	N/A	N/A	1.804–2.524	5.528	1.549	0.227–0.622
Residence Garden	0.325–0.388	0.296–1.288	2.254	N/A	7.168	1.17
School Playground	N/A	N/A	2.078	N/A	2.282–8.717	5.833
Sediment	0.431	1.256	16.66	2.792	2.515	N/A

## Data Availability

Restrictions apply to the availability of these data. Data are owned by the Ministry of Health of the Republic of Karakalpakstan (MOHK) and are available from the corresponding author with the permission of MOHK.

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
