# Peer review of "Organochlorine Pesticides and Salinity in Karakalpakstan, Uzbekistan: Environmental Health Risks Associated with the Aral Sea Crisis"

_ijerph, 2025, doi:10.3390/ijerph22111751_

Round 1

Reviewer 1 Report

Comments and Suggestions for Authors

Dear Authors,

The research presented is undoubtedly of great relevance to the region and to other areas facing environmental discrimination, inadequate management of agrochemicals, wastewater, and urban solid waste, as well as—most critically—the lack of sufficient and safe water for the population, which constitutes a fundamental human right. However, the current presentation of the manuscript does not align appropriately with the significance of the topic addressed. Therefore, I kindly request that you implement all necessary revisions, as these will provide the essential elements to substantiate the claims made, particularly in the discussion and conclusion sections. I look forward to reviewing your manuscript once these changes have been completed.

Comments on the Quality of English Language

It is essential that the grammar and English language style of the manuscript be thoroughly reviewed by a native English speaker

Reviewer 2 Report

Comments and Suggestions for Authors

Bartrem et al. investigated the presence of organochlorine pesticides and salinity in Karakalpakstan (Uzbekistan) and appraised the environmental health risk associated with the Aral Sea crisis. The work is of fine quality and will certainly contribute to the field. Moreover, this study will provide a foundation for future researchers in the area. I suggest the paper be accepted after incorporating the following minor comments.

  1. If possible, provide detailed protocols for samples’ preparation.
  2. Table 2 has symbols; it would be better to indicate what they represent in the table footnote (like table 7).
  3. It would be better to tabulate the air quality results under heading 3.4. But, if the data is not significant, it can still be provided as supplementary data (in supplementary materials) for future researchers.  

Reviewer 3 Report

Comments and Suggestions for Authors

The study is very important to many countries around the world, that have problems of water salination and environmental pollution due to pesticides. The authors are raising an important point about these two aspects in Karakalpakstan, Uzbekistan of elevated levels of salt in drinking water, and pesticides and carcinogen (hexavalent chromium), all of which have a historical connection.

Author have presented the methodology and the result very well. However, they failed to discuss the result well, which diminishes the strength of this manuscript. In the discussion, they included aspects that where not captured in the Methodology section (e.g., observations, people's sentiments, etc.), I have highlighted that limitation in the manuscript. Furthermore, authors have consistently started with literature and then include the results (not explicitly), which in most cases were not connected, thus making the reading very bad. In some cases, the literature was not vague/generic and not relevant.

Authors should read other people's work to see the style of writing, particularly the Discussion, conclusion and the abstract.

Should these aspects be addressed properly, the manuscript stands a better chance of being published as I believe it will make an interesting read and could add to the body of knowledge for researchers focussing on this area in Uzbekistan.

Round 2

Reviewer 3 Report

Comments and Suggestions for Authors

Thank you very much for improving the quality of the manuscript. 

There is only one comment that needs clarification, please fix the sampling duration, was it between March-April 2023 or March-April 2024?

Author Response

Comment 1: 

Please check the subsequent dates. they say March-April 2023. I have marked them.

Please make the necessary corrections.

Response: Thank you. The year should say 2023 and we have corrected this. 

Comment 2: Monograph IS aligned

Response: Thank you. We have made this correction.

All other comments are about the incorrect use of 2024, which we corrected. Thank you for catching this error. Sampling was done in 2023. 
